# Use of the Azure Kinect to measure foot clearance during obstacle crossing: A validation study

Kohei Yoshimoto, Masahiro Shinya*

Graduate School of Humanities and Social Science, Hiroshima University, Higashi-Hiroshima, Japan

* mshinya@hiroshima-u.ac.jp

## Abstract

Obstacle crossing is typical adaptive locomotion known to be related to the risk of falls. Previous conventional studies have used elaborate and costly optical motion capture systems, which not only represent a considerable expense but also require participants to visit a laboratory. To overcome these shortcomings, we aimed to develop a practical and inexpensive solution for measuring obstacle-crossing behavior by using the Microsoft Azure Kinect, one of the most promising markerless motion capture systems. We validated the Azure Kinect as a tool to measure foot clearance and compared its performance to that of an optical motion capture system (Qualisys). We also determined the effect of the Kinect sensor placement on measurement performance. Sixteen healthy young men crossed obstacles of different heights (50, 150, and 250 mm). Kinect sensors were placed in front of and beside the obstacle as well as diagonally between those positions. As indices of measurement quality, we counted the number of measurement failures and calculated the systematic and random errors between the foot clearance measured by the Kinect and Qualisys. We also calculated the Pearson correlation coefficients between the Kinect and Qualisys measurements. The number of measurement failures and the systematic and random error were minimized when the Kinect was placed diagonally in front of the obstacle on the same side as the trail limb. The high correlation coefficient (r > 0.890) observed between the Kinect and Qualisys measurements suggest that the Azure Kinect has excellent potential for measuring foot clearance during obstacle-crossing tasks.

## Introduction

Falls may have a serious impact on health, independence, and quality of life in the elderly population. Among elderly people over 75 years old, 24% of those who fell were severely injured, and 6% suffered fractures [1]. Tripping over obstacles is one of the most frequent cause of falls, reported that approximately 30 to 50% of falls among elderly people were caused by tripping [2, 3]. According to the systematic review by Galna et al. [4], research on effect of aging on obstacle-crossing behavior has been getting much attention. Understanding the mechanisms

**Data Availability Statement:** All relevant data are within the paper and its Supporting Information files.

**Funding:** This study was supported by JSPS/MEXT KAKENHI Grant-in-Aid, Grant Number: 17H04750,

21H05334. The funders had no role in study design, data collection and analysis, decision to publish, or preparation of the manuscript.

**Competing interests:** The authors have declared that no competing interests exist.

behind obstacle-crossing behavior in the interest of fall prevention is essential in an aging modern society.

Quantitative evaluation of gait parameters during obstacle negotiation, such as foot clearance (the vertical distance from the foot to the obstacle), is crucial in obstacle-crossing studies. The foot clearance is a versatile variable in fall prediction studies. Not only insufficient foot clearance (i.e., being short in absolute value), but large asymmetry or variability in foot clearance were also known as fall risk indicators. For example, it is suggested that an increased asymmetry in the foot clearance was related to the decline in physical function in the elderly [5–8]. Another previous study in healthy adults reported that greater variability in foot clearance is associated with a greater risk of contact with obstacles [9].

The most significant barriers to obstacle-crossing research include the high cost and lack of portability of optical motion capture systems. Markerless motion capture is a promising solution to overcome the weakness of optical motion capture systems that require reflecting markers. The Microsoft Kinect is a low-cost, portable device containing a body tracking system that can measure three-dimensional joint positions. The usefulness of the Kinect body tracking system has been validated in gait and posture studies and clinical settings [10]. In contrast to most gait and posture tasks such as regular walking and standing, measuring foot clearance during obstacle crossing tasks requires data on the position of an external object (i.e., the obstacle) in addition to the subject's skeleton. This means that the body skeleton data measured by the Kinect, which are in camera-centered coordinates, should be transformed into the global coordinate system that defines the location of the obstacle. Although one study used the Kinect v2 to measure foot clearance [11], the coordinate transformation of Kinect data has not yet been validated.

The effect of the location of the Kinect sensor also needs to be investigated. In gait analysis, researchers often placed the first- and second-generation Kinect sensors in front of the participant [12–16], while a few studies placed the sensor diagonally in front of the participant [17, 18]. A recent study showed that the optimal camera angle might differ depending on the version of Kinect sensor that is used [17]. Yeung et al. [17] recorded the lower limb joint angles of the Kinect side during treadmill walking with the Kinect v2 and the Azure Kinect at five camera angles: 0˚, 22.5˚, 45˚, 67.5˚, and 90˚. The results showed that the error was minimized at oblique rather than head-on camera angles. However, validation in the previous study was only conducted on the measurement of the unilateral kinematics because steady-state walking is a symmetrical movement.

Unlike the steady-state walking, researchers often record kinematics from the both of the lead and trail limbs. For example, the lower clearance in the trail limb compared to the lead limb was observed in older adults with cognitive impairment [5, 8]. To calculate the asymmetry of the foot clearance, systematic errors should be identical for the both sides. If one tries to record the kinematics from both sides by using a single Kinect sensor, self-occlusion could drastically impact the measurement quality. Seo et al. [19] reported that the contralateral Kinect location made a large error due to being occluded by other body parts. This means that the measurement performance could be different between the left and right sides of the body if the motion was captured from the Kinect placed at one side of the participant. In this study, we compared the measurement performance between the lead and trail limbs during obstacle crossing.

The purpose of this study was to validate the Azure Kinect for assessing foot clearance during obstacle crossing. We tested the effects of the Kinect sensor location and the obstacle height on the measurement of the foot clearance of the lead and trail limbs. To this end, obstacle-crossing behavior for obstacles of different heights was simultaneously recorded by a conventional motion capture system and the Azure Kinect sensors from four different viewing angles.

Then, we analyzed the measurement error as the difference between the foot clearances measured by the Azure Kinect sensor and the gold standard method (i.e., optical motion capture). Finally, the measurement errors were compared between the four Kinect locations to find the recommended Kinect placement.

## Methods

### Participants

Sixteen healthy young men (age: 21.6 ± 0.8 years; height: 170.7 ± 6.2 cm; weight: 59.4 ± 6.6 kg) participated in this study. The participants did not have any disability that could influence walking. The purpose and precautions of the study were fully explained to the participants in advance, and informed consent was obtained in written form from all of the participants. The study was conducted in accordance with the Declaration of Helsinki and approved by the local ethics committee at the Graduate School of Integrated Arts and Sciences, Hiroshima University (approval number: 01–31).

### Experimental setup

Participants walked barefoot at their own pace and crossed an obstacle. They were instructed to step over the obstacle with their right foot at the 7th step and then continue walking at least four more steps (Fig 1A). The participants practiced up to 3 trials before the recording trials for each obstacle condition. The obstacle was composed of a 1.00 m long plastic rod with a diameter of 10 mm across an aluminum frame. We used the hurdle-like obstacle because box-shaped obstacles occluded the foot from the Kinect sensors. Three obstacle heights were tested: 50, 150, and 250 mm. The obstacle heights were determined based on previous studies that used 51 mm and 152 mm [20] or 260 mm obstacle heights [21].

Whole-body kinematics was measured by two Azure Kinect sensors (Microsoft) placed at the height of 0.95 m and eight optical cameras (Miqus M3, Qualisys, Sweden). In this study, the Kinect sensors were set up in two different layouts. In one layout, two Kinect sensors were placed at 20˚ and 40˚ diagonal positions to the direction of the walking path on the trail limb side (location A and location B in Fig 1B). In another layout, Kinect sensors were placed at 75˚ and -75˚ angles, and images of the walking participants were captured from both the lead and trail limb sides (location C and location D in Fig 1B). Ten trials were recorded for each obstacle height and Kinect layout. Any trial in which the participant hit the obstacle was considered a failed trial, and the participant was allowed to repeat the trial until ten successful trials were recorded. The Azure Kinect Body Tracking SDK provided 3D skeleton data constructed of 32 body points at a sampling rate of 30 Hz (Sensor-SDK-v1.4.1, body-tracking-SDK-v1.0.1).

In our preliminary recordings and a previous study [22], it was observed that infrared markers placed on the ankle and knee for optical motion capture could hinder the recording of depth images by the Azure Kinect sensor. Instead of placing a marker on the knee and ankle, we created a rigid body model for the shank segment before the recording session. The rigid body model was created from markers attached to the medial and lateral femoral epicondyles and on the medial and lateral malleoli as well as two markers on the middle of the shank segment. When we recorded the obstacle-crossing task using the Kinect and Qualisys, we removed the markers on the medial or lateral femoral epicondyle and the medial or lateral malleolus, and the locations of these points were estimated from the other four markers of the shank rigid body (Fig 2). Except for the shank segments, significant interference of the infrared reflective markers with Qualisys recording was not observed. We placed markers on the left and right tragus, acromion, anterior superior iliac spine (ASIS), posterior superior iliac spine (PSIS), greater trochanter (GTR), first metatarsal bone, and radial and ulnar styloid processes.

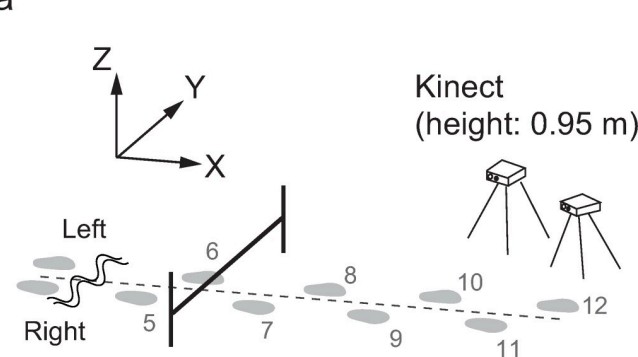

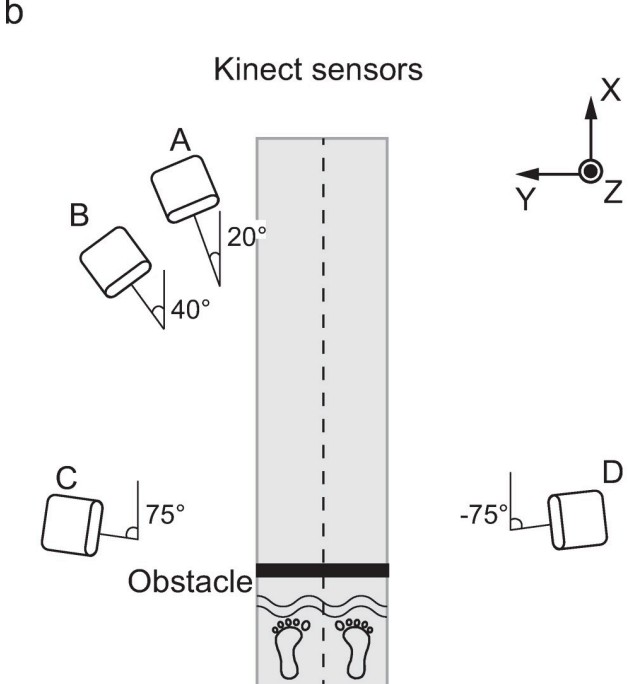

**Fig 1. Experimental diagram.** Participants crossed an obstacle (height: 50, 150, 250 mm) at the 7th step, with their right limb being the lead limb (a). The Kinect sensors were set up in two different layouts (b). In one layout, the sensor positions (location A and location B) were located at a 20˚ angle and 40˚ angle relative to the walkway on the trail limb side; in the other layout, the sensor positions (location C and location D) were located on opposite sides, at a 75˚ angle and a -75˚ angle.

## Data processing

The original skeleton data obtained from the Kinect SDK were in a coordinate system based on the depth camera of the Kinect. The skeleton data were transformed into the laboratory coordinate system for calculation of the foot clearance and the distance between the foot and obstacle. For this transformation, we carried out the following steps (Fig 3). 1) We placed two boards in the laboratory, and the point cloud data (resolution: 640×576) were collected from the Kinect depth image. 2) The plane of the floor was estimated using the Random Sample Consensus (RANSAC) algorithm in the MATLAB computer vision toolbox, and the tilt angle of the Kinect sensor was calculated. 3) After correction for the tilt angle of the Kinect, the

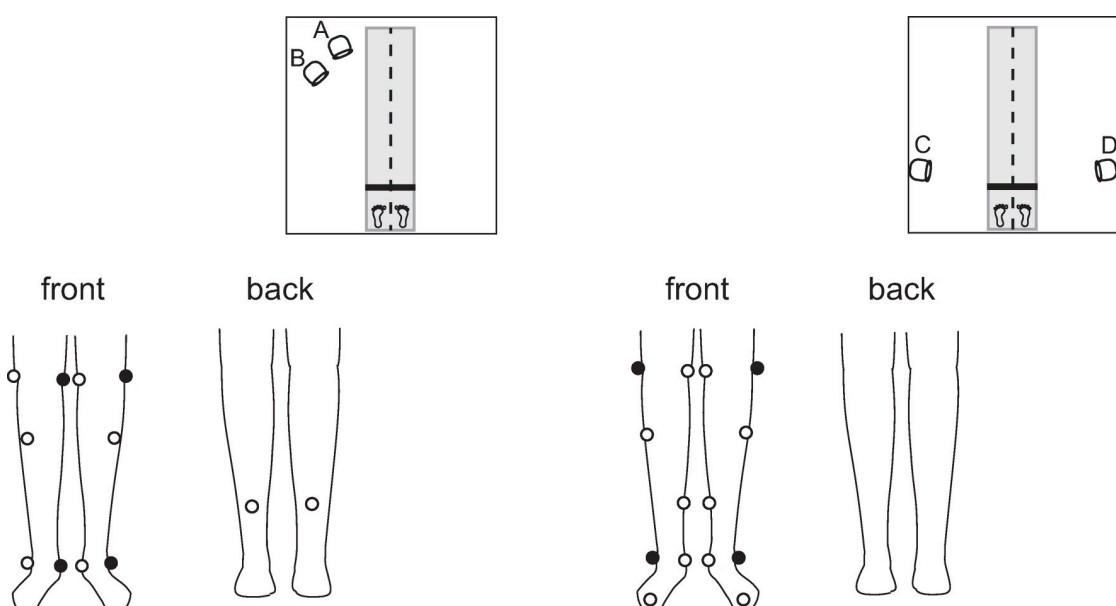

**Fig 2. Reflective marker placement.** A rigid body model was made for the shank segment from six infrared reflective markers. During the recording obstacle-crossing behavior, the markers on the side of the knee and ankle facing the Kinect were removed to minimize the interference between the Azure Kinect and the optical motion capture.

distance from the Kinect camera to the floor plane was calculated as the z coordinate of the center of gravity of the floor plane. 4) The plane of one of the two boards was estimated, and its tilt angle was calculated. 5) After correction for the tilt angle of the board plane, the coordinates of the intersection point on the line extending through the two boards were calculated. Once the transformation matrix from Kinect coordinates to laboratory coordinates was obtained, the skeleton data were transformed into laboratory coordinates.

Kinect measurements with a confidence level of high or medium were used, and those with low or no confidence were replaced with estimated values calculated by spline interpolation. Although the data with low or no confidence were discarded, apparently incorrect data were observed where the position of the toe marker was estimated to be external to the body. These trials were defined by multiple peaks with a significant estimated dropdown greater than 100 mm in the vertical displacement of the toe during obstacle crossing. Trials in which the foot trajectory was estimated to be lower than the obstacle were also regarded as erroneous trials. These errors in the Kinect measurements were considered measurement failures. We counted the number of measurement failures for each Kinect location, obstacle height, and limb (lead/trail). The number of measurement failures was used as an index of measurement quality. If the number of measurement failures exceeded 5 out of 10 trials for each condition, the data were excluded from the subsequent analysis of foot clearance.

The Kinect data and the Qualisys data were smoothed using a zero-lag second-order low-pass digital Butterworth filter with a cutoff frequency of 5 Hz. Spline interpolation was used to resample the Kinect data to 240 Hz from 30 Hz. Foot clearance was defined as the vertical distance from the toe to the obstacle when the toe was directly above the obstacle. The foot clearance calculated by the Qualisys was compensated by subtracting the vertical offset of the toe marker during standing. We also calculated residuals in the foot clearance by subtracting the Qualisys data from the Kinect data (Fig 4A). Then, we defined the systematic error as the within-subject mean value of the residuals and the random error as the within-subject standard deviation of the residuals (Fig 4B).

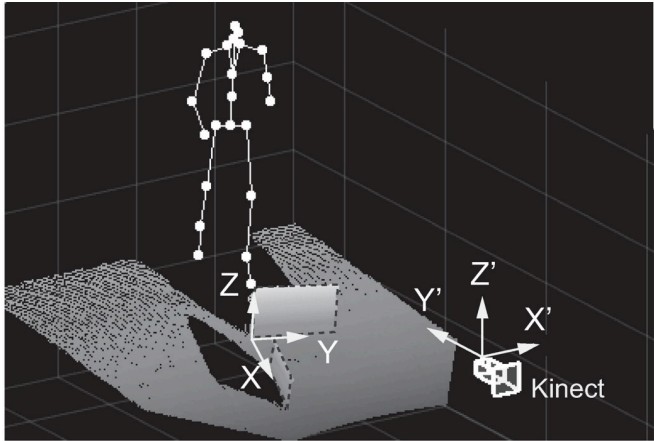

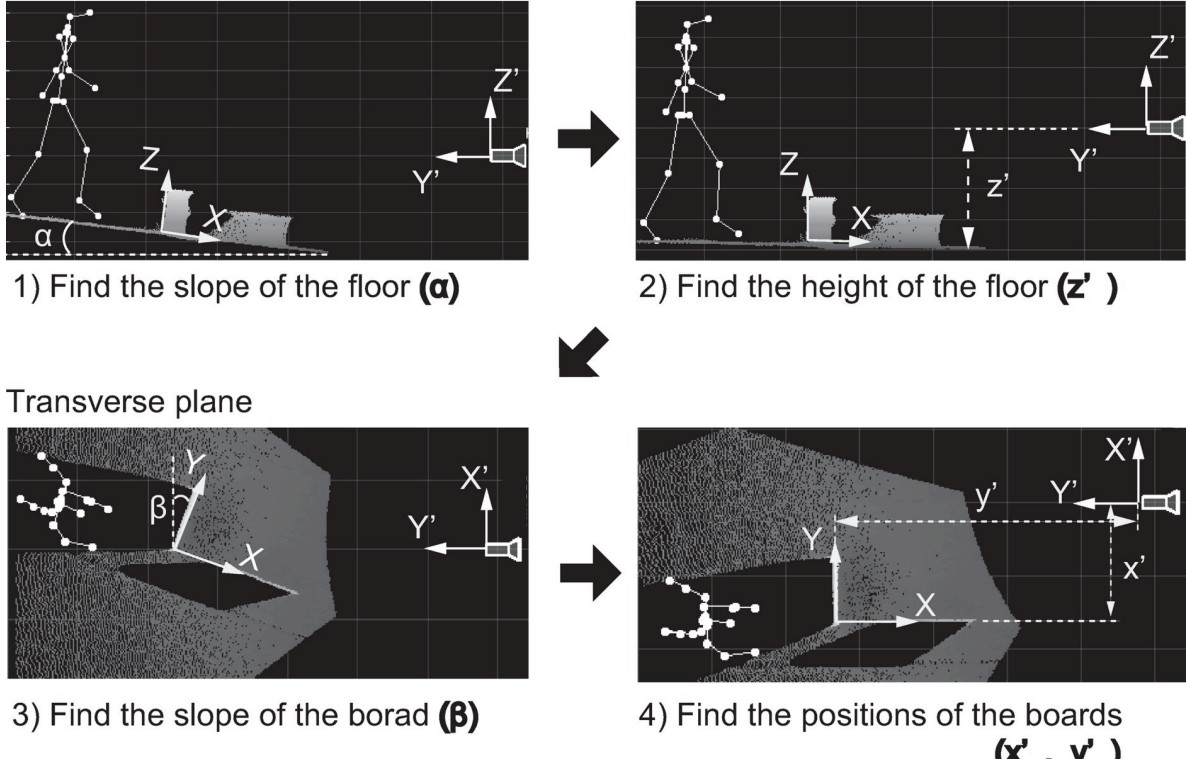

**Fig 3. Transformation from the Kinect's skeleton coordinates (X′, Y′, Z′) to laboratory coordinates (X, Y, Z).** Point cloud (resolution: 640×576) data were used for the transformation.

## Statistical analyses

Because there were unignorable inter-participant variability in the measurement performance and samples were not necessarily normally distributed, we used non-parametric statistical tests in the present study. Median [minimum, maximum] was used as descriptive. Friedman tests were used to evaluate differences in the number of measurement failures, the systematic error,

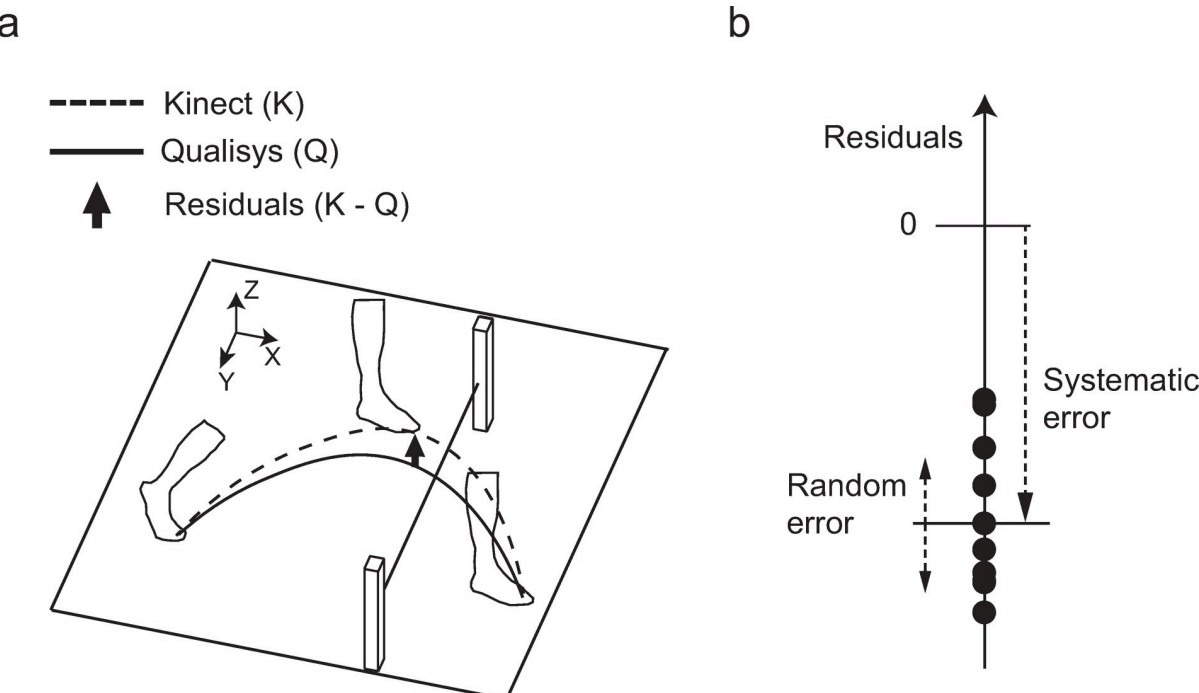

**Fig 4. Calculation of the systematic and random error.** Residuals were calculated by subtracting Qualisys from Kinect (a). The systematic error and random error were defined as the mean and standard deviation, respectively, of the residuals (b).

and the random error among Kinect locations. Kendall's W was used as an index of effect size for the Friedman tests. The Friedman tests were performed separately for the lead limb in the 50 mm obstacle condition, for the lead limb in the 150 mm obstacle height condition, for the lead limb in the 250 mm obstacle height condition, for the trail limb in the 50 mm obstacle condition, for the trail limb in the 150 mm obstacle height condition, and for the trail limb in the 250 mm obstacle height condition. If a significant main effect was observed in the Friedman tests, multiple comparisons were performed with Conover's test, and the significance level was adjusted by using the Holm correction (6 pairs).

A one-sample Wilcoxon signed-rank test was performed to determine whether the systematic error was significantly different from 0. Pearson's correlation analysis was performed between the within-subject average of the foot clearances measured by Kinect and that measured by Qualisys. As location B (i.e., oblique viewing angle) showed the best measurement performance (see Results for details), the systematic error and the random error were compared between limbs (lead, trail) and obstacle heights (50, 150, 250 mm) by using pairwise Wilcoxon signed-rank tests with a significance level adjusted by using the Holm correction (15 pairs). The significance level was set to 0.05.

## Results

When we describing the results of the Friedman tests, we use abbreviations for mentioning the limb and the obstacle height conditions: the lead limb in the 50 mm obstacle condition (L50), the trail limb in the 250 mm obstacle condition (T250), etc. Friedman tests revealed that the number of measurement failures was different depending on the Kinect location except for the L50 measurement (Table 1). According to Conover's multiple comparison tests, significantly fewer measurement failures occurred in location B than in locations C and D for the L150

measurement. Locations A and B showed significantly fewer measurement failures than location C for the L250 measurement. In T50 and T150, locations A, B, and C showed significantly fewer measurement failures than location D. In T250, location C showed significantly fewer measurement failures than locations A and D.

If the number of measurement failures exceeded 5 out of 10 trials for each condition for a participant, the data of the participants were excluded from the subsequent analysis of foot clearance. For example, n = 3 for L250 (Table 1) means that 13 participants were excluded from the following analysis because of this criterion. The systematic error of the foot clearance was also influenced by the Kinect location. Friedman tests showed significant main effects of the Kinect location for L50, L150, T50, T150, and T250. There was no significant main effect observed for L250 because the degree of freedom for the test was very small due to the large number of measurement failures. Given that the zero systematic error is the best performance, the Kinect measurement from location B showed a greater agreement with the Qualisys measurement than the measurements from locations A and C for L50. In the L150 condition, there were significant differences in the systematic error between the Kinect location A and B, between B and C, and between C and D. In the T50, significant differences were observed between location A and B, between B and D, and between C and D. In the T150, there were significant differences between location A and B, and between B and D. In the T250, there were significant differences between location A and B, and between A and C. A one-sample Wilcoxon signed-rank test revealed that the systematic error of the foot clearance in the lead limb was significantly less than 0 for all the Kinect locations and obstacle heights tested (Table 2). The systematic error of the foot clearance in the trail limb in the measurement from

**Table 1. Comparison of foot clearance measurement performance between Kinect locations.**

| | | Kinect location | | | | | | | | Friedman test | | | | Multiple comparison |
|---|---|---|---|---|---|---|---|---|---|---|---|---|---|---|
| | | A | | B | | C | | D | | $X^2_{(3)}$ | n | p | Kendall's W | |
| number of measurement failures | | | | | | | | | | | | | | |
| lead | 50 | 0.0 | [0, 4] | 0.0 | [0, 0] | 0.0 | [0, 2] | 0.0 | [0, 3] | 4.368 | 16 | 0.224 | 0.091 | |
| | 150 | 0.0 | [0, 1] | 0.0 | [0, 0] | 0.0 | [0, 7] | 0.5 | [0, 10] | 12.708 | 16 | 0.005* | 0.265 | B < C, B < D |
| | 250 | 0.0 | [0, 1] | 0.0 | [0, 2] | 4.5 | [0, 10] | 1.0 | [0, 10] | 26.476 | 16 | < 0.001* | 0.552 | A < C, B < C |
| trail | 50 | 0.0 | [0, 2] | 0.0 | [0, 0] | 0.0 | [0, 0] | 3.0 | [0, 7] | 37.971 | 16 | < 0.001* | 0.791 | A < D, B < D, C < D |
| | 150 | 0.0 | [0, 1] | 0.0 | [0, 0] | 0.0 | [0, 0] | 5.5 | [0, 9] | 41.727 | 16 | < 0.001* | 0.869 | A < D, B < D, C < D |
| | 250 | 3.0 | [1, 8] | 1.0 | [0, 7] | 0.0 | [0, 1] | 3.0 | [0, 10] | 21.579 | 16 | < 0.001* | 0.450 | A > C, C < D |
| systematic error | | | | | | | | | | | | | | |
| lead | 50 | -29.8 | [-42.3, -10.0] | -11.2 | [-51.6, -0.8] | -34.2 | [-70.6, -3.3] | -21.1 | [-150.7, 2.2] | 16.500 | 16 | < 0.001* | 0.344 | A < B, B > C |
| | 150 | -27.8 | [-62.8, -9.3] | -13.7 | [-53.5, -2.9] | -50.7 | [-89.7, 0.0] | -20.00 | [-122.2, 5.2] | 20.908 | 13 | < 0.001* | 0.536 | A < B, B > C, C < D |
| | 250 | -28.2 | [-93.9, -4.6] | -19.9 | [-57.4, 0.0] | -68.9 | [-111.8, -18.9] | -26.4 | [-51.8, 12.0] | 6.600 | 3 | 0.086 | 0.733 | |
| trail | 50 | -33.9 | [-84.4, 2.3] | 6.4 | [-12.7, 17.7] | -2.0 | [-27.3, 11.0] | -104.0 | [-143.8, -0.1] | 31.114 | 14 | < 0.001* | 0.741 | A < B, B > D, C > D |
| | 150 | -37.7 | [-95.0, -0.4] | 8.0 | [-13.7, 22.2] | -2.8 | [-43.2, 15.1] | -37.2 | [-75.0, 89.2] | 16.650 | 8 | < 0.001* | 0.694 | A < B, B > D |
| | 250 | -62.1 | [-146.8, 26.1] | 15.6 | [-54.8, 30.0] | -0.9 | [-77.4, 24.2] | -50.0 | [-105.5, 12.6] | 20.600 | 9 | < 0.001* | 0.763 | A < B, A < C |
| random error | | | | | | | | | | | | | | |
| lead | 50 | 6.3 | [4.0, 11.3] | 5.5 | [3.8, 15.7] | 18.5 | [7.3, 36.5] | 9.8 | [4.1, 76.5] | 17.775 | 16 | < 0.001* | 0.370 | A < C, B < C |
| | 150 | 4.8 | [2.3, 16.4] | 6.8 | [2.4, 15.5] | 17.1 | [3.6, 33.6] | 10.0 | [3.4, 68.6] | 13.523 | 13 | 0.004* | 0.347 | A < C, B < C |
| | 250 | 6.6 | [2.7, 27.4] | 9.4 | [4.2, 16.2] | 36.9 | [8.5, 132.0] | 10.9 | [5.3, 48.3] | 5.400 | 3 | 0.145 | 0.600 | |
| trail | 50 | 13.2 | [6.7, 45.9] | 6.7 | [3.2, 11.7] | 8.3 | [3.1, 22.7] | 44.0 | [26.7, 136.3] | 34.543 | 14 | < 0.001* | 0.822 | A > B, B < D, C < D |
| | 150 | 15.9 | [5.2, 31.5] | 6.3 | [3.1, 11.6] | 6.9 | [3.3, 38.1] | 29.5 | [11.8, 163.3] | 15.450 | 8 | 0.001* | 0.644 | A > B, B < D |
| | 250 | 38.4 | [8.8, 58.3] | 12.4 | [2.9, 48.1] | 8.8 | [6.3, 71.9] | 28.4 | [19.8, 50.7] | 9.933 | 9 | 0.019* | 0.368 | |

Median [minimum, maximum] values were used as descriptive statistics for the number of measurement failures, systematic error, and random error.

**Table 2. Results of one-sample Wilcoxon signed-rank tests for the systematic error.**

| | | Kinect location | | | | | | | | | | | |
|---|---|---|---|---|---|---|---|---|---|---|---|---|---|
| | | A | | | B | | | C | | | D | | |
| systematic error | | | | | | | | | | | | | |
| lead | 50 | -29.8 | [-42.3, -10.0] | p < 0.001 | -11.2 | [-51.6, -0.8] | p < 0.001 | -34.2 | [-70.6, -3.3] | p < 0.001 | -21.1 | [-150.7, 2.2] | p < 0.001 |
| | 150 | -27.8 | [-62.8, -9.3] | p < 0.001 | -13.7 | [-53.5, -2.9] | p < 0.001 | -50.7 | [-89.7, 0.0] | p < 0.001 | -20.00 | [-122.2, 5.2] | p < 0.001 |
| | 250 | -28.2 | [-93.9, -4.6] | p < 0.001 | -19.9 | [-57.4, 0.0] | p < 0.001 | -68.9 | [-111.8, -18.9] | p = 0.004 | -26.4 | [-51.8, 12.0] | p = 0.027 |
| trail | 50 | -33.9 | [-84.4, 2.3] | p < 0.001 | 6.4 | [-12.7, 17.7] | p = 0.105 | -2.0 | [-27.3, 11.0] | p = 0.404 | -104.0 | [-143.8, -0.1] | p < 0.001 |
| | 150 | -37.7 | [-95.0, -0.4] | p < 0.001 | 8.0 | [-13.7, 22.2] | p = 0.004 | -2.8 | [-43.2, 15.1] | p = 0.404 | -37.2 | [-75.0, 89.2] | p = 0.195 |
| | 250 | -62.1 | [-146.8, 26.1] | p < 0.001 | 15.6 | [-54.8, 30.0] | p = 0.303 | -0.9 | [-77.4, 24.2] | p = 0.782 | -50.0 | [-105.5, 12.6] | p = 0.020 |

Median [minimum, maximum] values were used as descriptive statistics for the systematic error. The alternative hypothesis specifies that the median is different from 0.

locations B and C was not significantly different from 0 except for the 150 mm obstacle condition measured from location B. The systematic error of the foot clearance in the trail limb in the measurement from locations A and D was significantly different from 0 except for the 150 mm obstacle condition measured from location D.

For the random error, Friedman tests showed significant main effects of the Kinect location for L50, L150, T50, T150, and T250. There was no significant main effect observed for L250. In the L50 and L150 conditions, significantly smaller random errors were observed in locations A and B than in location C. In T50, significant differences were observed between locations A and B, between B and D, and between C and D. In the T150 condition, significantly smaller random error was observed in location B than in location A and D.

Correlations between the foot clearances measured by the Kinect and Qualisys are illustrated in Fig 5. Overall, the Kinect measurements from location B showed good agreement with the Qualisys measurements for both the lead and trail limbs (r > 0.890).

The systematic and random error of the foot clearance in the measurement from location B were compared between limbs and obstacle height conditions (Table 3). Significantly smaller systematic errors were observed in the lead limb than in the trail limb. In the comparisons between obstacle height conditions, there was a significant difference in the systematic error between L50 and L150, and between L150 and L250. The random error in L50 was significantly smaller than that in T250.

## Discussion

In this study, we validated Azure Kinect sensors for the measurement of foot clearance during an obstacle-crossing task. The results suggest that foot clearance was reliably measured by the Azure Kinect sensor and that the sensor should be placed diagonally in front of the obstacle on the same side as the trail limb (location B in this study), which is consistent with a previous study [17]. The number of measurement failures was larger when the Kinect was placed by the side of the obstacle (locations C and D) than when it was placed directly or diagonally in front (locations A and B). Contralateral measurements (i.e., the lead limb measured from location C and the trail limb measured from location D) had more measurement failures, larger systematic error, and larger random error than ipsilateral measurements, perhaps because the closer occluded the farther limb [18, 19] (typical depth images are shown in S1 Fig). Notably, the sensors that were placed beside the obstacle did not perform well even for ipsilateral measurements (typical depth image shown in S2 Fig).

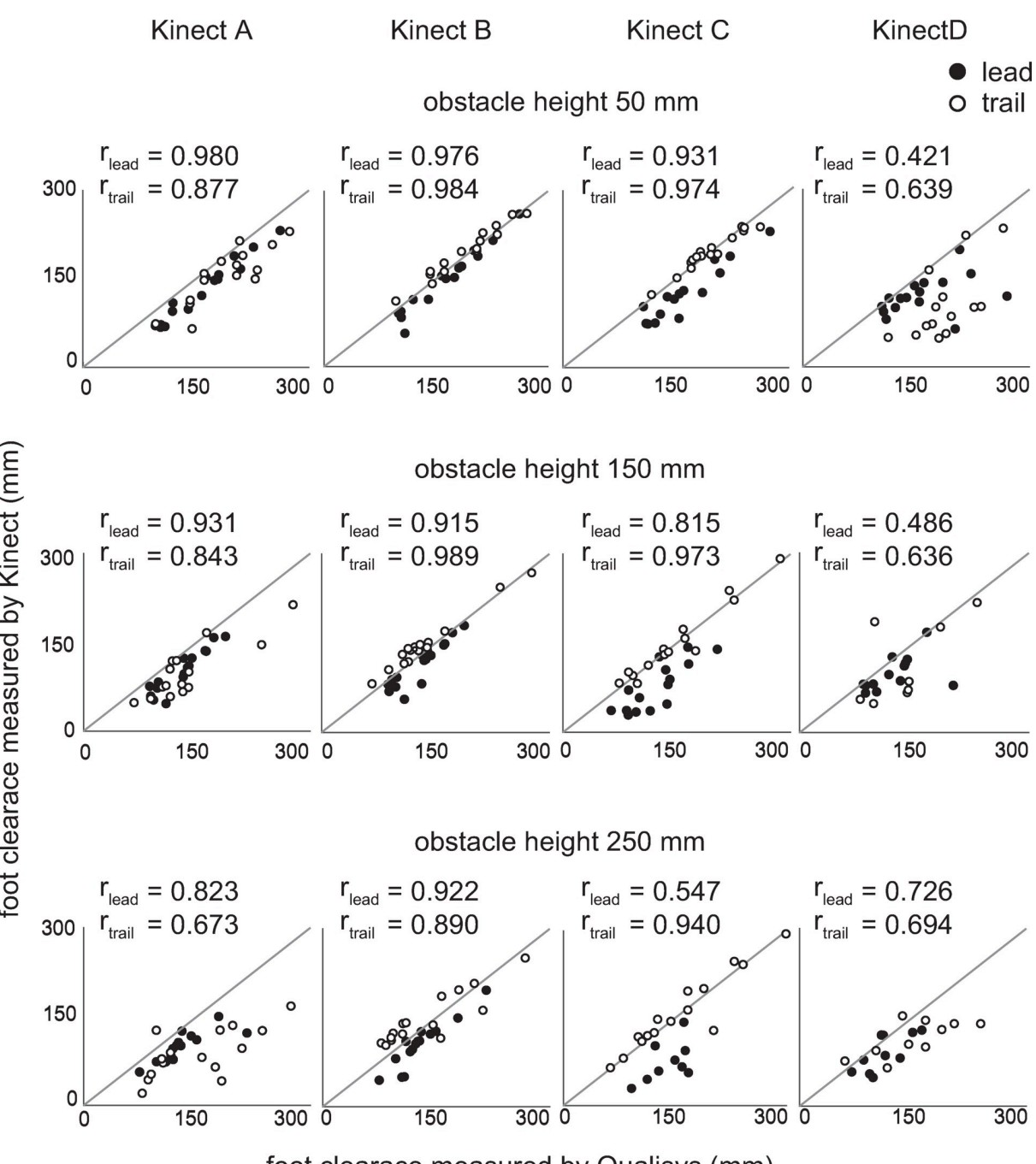

**Fig 5. Relationship between the foot clearance measured by the Kinect and Qualisys.** The diagonal straight lines represent the identity line, y = x. The black circles represent the data for the lead limb, and the white circles represent those for the trail limb. Pearson's correlation coefficients were calculated separately for the lead and trail limbs.

Based on the systematic and random error, Kinect location B showed better performance than any other tested placement. Correlation analysis also suggested that the foot clearance measured by a single diagonally placed Azure Kinect was reliable for both the lead and trail limbs. The observed poor performance of the sagittally placed Kinects might be due to self-occlusion, as discussed in the previous paragraph. The position in front of the walking subject

**Table 3. Systematic error and random error measured in location B were compared between limbs and obstacle height conditions by using Wilcoxon signed-rank tests.**

|  | obstacle height | lead | | trail | |
|---|---|---|---|---|---|
| systematic error | 50 | -11.2 | [-51.6, -0.8]* | 6.4 | [-12.7, 17.7]# |
|  | 150 | -13.7 | [-53.5, -2.9]* | 8.0 | [-13.7, 22.2]# |
|  | 250 | -19.9 | [-57.4, 0.0] | 15.6 | [-54.8, 30.0]# |
| random error | 50 | 5.5 | [3.8, 15.7] | 6.7 | [3.2, 11.7]* |
|  | 150 | 6.8 | [2.4, 15.5] | 6.3 | [3.1, 11.6]* |
|  | 250 | 9.4 | [4.2, 16.2] | 12.4 | [2.9, 48.1] |

Median [minimum, maximum] values were used as descriptive statistics for the systematic error and random error. Holm correction's significant difference *: $p < 0.05$ (vs. 250 mm obstacle height);

#: $p < 0.05$ (Lead vs. Trail)

(location A) was also not the optimal sensor location. The systematic and random error were larger for location A than for location B. This might be because the oblique camera view angle provides information about both the sagittal and frontal planes [17].

The poor measurement performance of the Kinect in front of the subject was significant for the trail limb crossing the high (250 mm) obstacle. This was probably because the trail limb, unlike the lead limb, crossed over the high obstacle by flexing the knee joint rather than the hip and ankle joints [23]. This knee flexion hides the foot segment behind the thigh segment, leading to unreliable estimation of the toe position during obstacle crossing.

As discussed above, the Azure Kinect can be a reliable portable solution for measuring foot clearance during obstacle-crossing tasks. Previous studies with older versions of the Kinect reported that toe markers generated a large amount of noise [16, 24]. In contrast, a recent study reported that the Azure Kinect had improved toe tracking performance compared to the Kinect v2 [25]. Additionally, the depth sensor of the Azure Connect has a higher resolution (640×576) than that of the Kinect v2 (512×424), which may contribute to the reduction of the error in toe positioning.

Caution should be taken in drawing within-subject comparisons of foot clearance between the lead and trail limbs or between obstacle heights. The foot clearance of the lead limb was underestimated by approximately 10 to 20 mm, suggesting that one should be careful when assessing asymmetry of the clearance by using a single Azure Kinect sensor, for example. The measurement performance was relatively poor for the 250 mm obstacle height condition, possibly because of the self-occlusion of the foot segment due to the knee flexion motion. As a limitation of the present study, we did not test the effect of the height of the Kinect sensors. Also, as we only recruited healthy young adults participants, the measurement performance should be validated for those with different gait characteristics such as the elderly population or neuromuscular patients [26].

## Supporting information

**S1 Fig. Depth image in contralateral measurements.**
(TIF)

**S2 Fig. Depth image in ipsilateral measurements.**
(TIF)

**S1 Table. Within-participant number of measurement failures for each limb and obstacle height.**
(XLSX)

**S2 Table. Within-participant systematic error of foot clearance for each limb and obstacle height.**
(XLSX)

**S3 Table. Within-participant random error of foot clearance for each limb and obstacle height.**
(XLSX)

## Acknowledgments

We would like to thank all participants in the present study.

## Author Contributions

**Conceptualization:** Masahiro Shinya.

**Data curation:** Kohei Yoshimoto, Masahiro Shinya.

**Formal analysis:** Kohei Yoshimoto, Masahiro Shinya.

**Funding acquisition:** Masahiro Shinya.

**Investigation:** Kohei Yoshimoto, Masahiro Shinya.

**Methodology:** Kohei Yoshimoto.

**Project administration:** Masahiro Shinya.

**Resources:** Masahiro Shinya.

**Supervision:** Masahiro Shinya.

**Visualization:** Masahiro Shinya.

**Writing – original draft:** Kohei Yoshimoto.

**Writing – review & editing:** Kohei Yoshimoto, Masahiro Shinya.

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
