## [Decision Letter · Decision Letter 0]

11 Feb 2022

PONE-D-21-33562Use of the Azure Kinect to measure foot clearance during obstacle crossing: A validation studyPLOS ONE

Dear Dr. Shinya,

Thank you for submitting your manuscript to PLOS ONE. After careful consideration, we feel that it has merit but does not fully meet PLOS ONE’s publication criteria as it currently stands. Therefore, we invite you to submit a revised version of the manuscript that addresses the points raised during the review process.

We look forward to receiving your revised manuscript.

Kind regards,

Shazlin Shaharudin

Academic Editor

PLOS ONE

Journal Requirements:

This study was supported by JSPS KAKENHI Grant Number 17H04750.

Reviewers' comments:

Reviewer's Responses to Questions

**Comments to the Author**

1. Is the manuscript technically sound, and do the data support the conclusions?

Reviewer #1: Yes

Reviewer #2: Yes

2. Has the statistical analysis been performed appropriately and rigorously? 

Reviewer #1: Yes

Reviewer #2: Yes

3. Have the authors made all data underlying the findings in their manuscript fully available?

Reviewer #1: No

Reviewer #2: Yes

4. Is the manuscript presented in an intelligible fashion and written in standard English?

Reviewer #1: Yes

Reviewer #2: Yes

5. Review Comments to the Author

Reviewer #1: The authors have reported data from kinect system reporting it's validity to capture foot clearance during obstacle navigation. Some of the minor concerns are as follows:

Line 49: do you mean risk of fall while contact with obstacles? Also, contact with obstacles does not necessarily imply that the strategy has to be crossing the obstacles. Someone could just avoid crossing the obstacle and take a different route. So, overall, in the introduction, please build a strong argument regarding clinical or daily importance of obstacle crossing. Currently, it is lacking.

Also, asymmetry in foot clearance is different than insufficient foot clearance which might be what you mean? So, please use the correct verbiage and reference in your context.

Line 51: You talk about variability which is a totally different biomechanical measure than asymmetry or clearance. May be you could put some more thought into the variables which are part of your manuscript and also related to the context of the introduction.

Line 70: based on this, would not you propose just using the camera diagonally for your study if diagonally has already been shown to be better? Yes/no, and why needs to be added to your introduction section.

Lines 72-78: seem thoughts have bene left incompletely addressed. Please reshape this paragraph.

Line 79: If the objective was foot clearance then why did you focus that you wanted to compare lead vs trail limb? And if you wanted to compare lead vs trail limb, then you need to state the reason and scientific and/or clinical reasoning for it.

Line 111: was there any familiarization trial? Yes/No and why? Pl add this information.

Line 166: how many these happened in your study?

Reviewer #2: Reviewer finds the article to be reasonably written.

Line 204 . Suggest to reintroduce all the acronyms i.e. L50, T250 in results

Line 227 . Table 1; perhaps this table can be simplified or organised in a way that shows the main significance?

6. PLOS authors have the option to publish the peer review history of their article (what does this mean?). If published, this will include your full peer review and any attached files.

Reviewer #1: No

Reviewer #2: **Yes: **Raihana Sharir

---

## [Author Response · Author response to Decision Letter 0]

20 Feb 2022

Dr. Shazlin Shaharudin

Editor-in-Chief, PLOS ONE

shazlin@usm.my

February 21, 2022

Ref.: PONE-D-21-33562

Use of the Azure Kinect to measure foot clearance during obstacle crossing: A validation study

Dear Dr. Shazlin Shaharudin

Thank you very much for editing our manuscript, “Use of the Azure Kinect to measure foot clearance during obstacle crossing: A validation study”, and the valuable comments of the two reviewers. We submit our revised manuscript, as well as a point-by-point response to the reviewers’ comments.

According to a comment from the academic editor of the journal, we have modified the statement of financial disclosure as follows. Also, we have added another funding source because we would pay the publishing fee from the grant (21H05334).

This study was supported by JSPS/MEXT KAKENHI Grant-in-Aid, Grant Number: 17H04750, 21H05334. The funders had no role in study design, data collection and analysis, decision to publish, or preparation of the manuscript.

We feel that the revised manuscript is a suitable response to the comment and is significantly improved over the initial submission. 

Thank you in advance for your kind consideration of this paper.

Sincerely yours,

Masahiro SHINYA

Hiroshima University

1-7-1 Kagamiyama, Higashi-Hiroshima, 739-8521, Japan

+81-82-424-4544

mshinya@hiroshima-u.ac.jp

Academic editor:

We have confirmed that our manuscript meets PLOS ONE’s style requirement.

This study was supported by JSPS KAKENHI Grant Number 17H04750.

We will change our financial disclosure.

“This study was supported by JSPS/MEXT KAKENHI Grant-in-Aid, Grant Number: 17H04750, 21H05334. The funders had no role in study design, data collection and analysis, decision to publish, or preparation of the manuscript.”

We have confirmed that our reference list is complete and correct.

We used PACE to check our figure files met the PLOS requirements, and revised the wasted space in Fig. 2 and Fig. 4.

 

Reviewer #1: 

The authors have reported data from kinect system reporting it's validity to capture foot clearance during obstacle navigation. Some of the minor concerns are as follows:

Thank you very much for your review. We revised the manuscript according to your and the other reviewer’s comments. In the revised manuscript, the modification associated with your comments are highlighted in green, and those with the other reviewer are highlighted in blue. Comment-by-comment responses are as follows.

Line 49: do you mean risk of fall while contact with obstacles? Also, contact with obstacles does not necessarily imply that the strategy has to be crossing the obstacles. Someone could just avoid crossing the obstacle and take a different route. So, overall, in the introduction, please build a strong argument regarding clinical or daily importance of obstacle crossing. Currently, it is lacking.

We strengthened the Introduction by adding detailed description and references.

“Tripping over obstacles is one of the most frequent cause of falls, reported that approximately 30 to 50 % of falls among elderly people were caused by tripping [2,3]. According to the systematic review by Galna et al. [4], research on effect of aging on obstacle-crossing behavior has been getting much attention.” (line 43-46)

Also, asymmetry in foot clearance is different than insufficient foot clearance which might be what you mean? So, please use the correct verbiage and reference in your context.

Line 51: You talk about variability which is a totally different biomechanical measure than asymmetry or clearance. May be you could put some more thought into the variables which are part of your manuscript and also related to the context of the introduction.

As you mentioned, insufficient foot clearance might be related to an increased risk of tripping. However, the relevance of the foot clearance is not limited to its raw value, fall risk indicators might be obtained by processing foot clearance. Actually, researchers have calculated the asymmetry or variability in foot clearance and shown that these were related to the fall risk. To make this point clear, we added the following sentences.

“The foot clearance is a versatile variable in fall prediction studies. Not only insufficient foot clearance (i.e., being short in absolute value), but large asymmetry or variability in foot clearance were also known as fall risk indicators.” (line 49-51)

Line 70: based on this, would not you propose just using the camera diagonally for your study if diagonally has already been shown to be better? Yes/no, and why needs to be added to your introduction section.

To emphasize the differences between the previous and the present studies, we added the following description.

“However, validation in the previous study was only conducted on the measurement of the unilateral kinematics because steady-state walking is a symmetrical movement.” (line 72-74)

Lines 72-78: seem thoughts have bene left incompletely addressed. Please reshape this paragraph.

Line 79: If the objective was foot clearance then why did you focus that you wanted to compare lead vs trail limb? And if you wanted to compare lead vs trail limb, then you need to state the reason and scientific and/or clinical reasoning for it.

We have made a substantial revision on this paragraph. In this paragraph, we firstly state the importance of recording both sides in obstacle crossing tasks. Then, we point out the potential self-occlusion problem, and finally, we briefly explain the comparison in the present study.

“Unlike the steady-state walking, researchers often record kinematics from the both of the lead and trail limbs. For example, the lower clearance in the trail limb compared to the lead limb was observed in older adults with cognitive impairment [5,8]. To calculate the asymmetry of the foot clearance, systematic errors should be identical for the both sides. If one tries to record the kinematics from both sides by using a single Kinect sensor, self-occlusion could drastically impact the measurement quality. Seo et al. [19] reported that the contralateral Kinect location made a large error due to being occluded by other body parts. This means that the measurement performance could be different between the left and right sides of the body if the motion was captured from the Kinect placed at one side of the participant. In this study, we compared the measurement performance between the lead and trail limbs during obstacle crossing.” (line 75-83)

Line 111: was there any familiarization trial? Yes/No and why? Pl add this information.

“The participants practiced up to 3 trials before the recording trials for each obstacle condition.” (line 105)

Line 166: how many these happened in your study?

The numbers of participants analyzed (i.e., not-excluded participants) were shown in Table 1. We added the explanation in the top of the paragraph describing the errors in foot clearance.

“If the number of measurement failures exceeded 5 out of 10 trials for each condition for a participant, the data of the participants were excluded from the subsequent analysis of foot clearance. For example, n = 3 for L250 (Table 1) means that 13 participants were excluded from the following analysis because of this criterion.” (line 239-242)

---

## [Editor Report · Decision Letter 1]

28 Feb 2022

Use of the Azure Kinect to measure foot clearance during obstacle crossing: A validation study

PONE-D-21-33562R1

Dear Dr. Shinya,

We’re pleased to inform you that your manuscript has been judged scientifically suitable for publication and will be formally accepted for publication once it meets all outstanding technical requirements.

Kind regards,

Shazlin Shaharudin

Academic Editor

PLOS ONE
---

## [Editor Report · Acceptance letter]

2 Mar 2022

PONE-D-21-33562R1 

Use of the Azure Kinect to measure foot clearance during obstacle crossing: A validation study 

Dear Dr. Shinya:

I'm pleased to inform you that your manuscript has been deemed suitable for publication in PLOS ONE. Congratulations! Your manuscript is now with our production department. 

Kind regards, 

on behalf of

Dr. Shazlin Shaharudin 

Academic Editor

PLOS ONE